# Utilization of Aflatoxin-B1-Contaminated Corn by Yellow Mealworm Larvae for Common Carp Feed and Assessing Residual Frass Toxicity by Zebrafish Embryo Microinjection

**DOI:** 10.3390/ijms26209851

**Published:** 2025-10-10

**Authors:** Zoltán Vajnai, Zsolt Csenki-Bakos, Balázs Csorbai, Tamás Bartucz, Illés Bock, Endre Csókás, Mátyás Cserháti, Balázs Kriszt, István Szabó

**Affiliations:** 1Department of Environmental Toxicology, Institute of Aquaculture and Environmental Safety, Hungarian University of Agriculture and Life Sciences (MATE), Páter Károly utca 1, 2100 Gödöllő, Hungary; vajnai.zoltan@phd.uni-mate.hu (Z.V.); csenki-bakos.zsolt.imre@uni-mate.hu (Z.C.-B.);; 2Department of Aquaculture, Institute of Aquaculture and Environmental Safety, Hungarian University of Agriculture and Life Sciences (MATE), Páter Károly utca 1, 2100 Gödöllő, Hungary; csorbai.balazs@uni-mate.hu (B.C.);; 3Department of Molecular Ecology, Institute of Aquaculture and Environmental Safety, Hungarian University of Agriculture and Life Sciences (MATE), Páter Károly utca 1, 2100 Gödöllő, Hungary; 4Department of Environmental Safety, Institute of Aquaculture and Environmental Safety, Hungarian University of Agriculture and Life Sciences (MATE), Páter Károly utca 1, 2100 Gödöllő, Hungary

**Keywords:** aflatoxin-B1-contaminated corn, mealworms, fish feed, common carp, zebrafish toxicity, microinjection

## Abstract

The aim of our study was to make one step further to verify a method that can turn back mycotoxin-contaminated crops into the circular economy. Thus, the possibility of utilizing aflatoxin B1 (AfB1)-contaminated corn by yellow mealworms (*Tenebrio molitor*) was investigated to be used as fish feed components. Four different self-contaminated corn samples were used in our study, of which one was below and three were above the threshold limit (20 µg/kg) regulated by the European Union. The highest applied AfB1 concentration in our study for insect feeding was 415 µg/kg (more than twenty times higher than the threshold). After a five-week feeding period insect mortality was not increased, even in the highly contaminated group, compared to the negative control. The mycotoxin in the dried and ground insects was only detected in the case of feeding with the highest-concentration corn, however it remained as low as 2.2 µg/kg. For studying the possible physiology effects, insect grounds were used in feeding experiments of common carp (*Cyprinus carpio*) fries. Results showed that insect meal, even if originated from a highly mycotoxin-contaminated crop, did not have a significant effect on the examined fish fries, compared with the control groups. The AfB1 concentrations of the leftover frass after insect rearing were also measured, and in the case of the highest concentration mealworm group, it was 157.6 µg/kg (other groups were under 20 µg/kg). Toxicity of frass extracts from different contaminated groups was also studied using microinjected zebrafish (*Danio rerio*) embryos. Extracts of the highly contaminated frass samples caused 91.67 ± 3.33% mortality and led to numerous phenotypic changes, which highlights the need for responsible usage of the by-product. However, the effects of injected frass samples, originating from corn with lower and more environmentally relevant AfB1 concentrations, were significantly lower.

## 1. Introduction

Aflatoxin B1 (AfB1) contamination is an increasing menace in Europe as climate chang creates better and better conditions for certain toxin-producing fungi species [1,2,3] Crops with mycotoxin levels highly exceeding thresholds are considered as hazardous wastes, meaning that they need to be eliminated somehow. This is a loss in the yield for farmers, even creating an expense through the cost of the treatment, or leading to illegal disposal. Countless studies have been analyzing the emerging number of mycotoxin exposures. In Europe the summers of 2022 and 2024 were times of drought leading to considerable amounts of mycotoxin-contaminated crops which drove the scientific community to study the diverse aspects of this scenario.

Climate change causes a tendency of highly mycotoxin-effected growing seasons occurring more and more often [4,5] which predicts no positive shift for the future. Furthermore, the area likely to be affected by the mycotoxin threat is expanding and spreading north year by year [1,6]. The whole verticum of agriculture is affected by the issue, meaning that not only is direct application of the contaminated grains problematic but mycotoxins are transmitted further in the food industry and detectable in dairy and brewing industries [7,8]. Even though there is an aspiration for working out advanced techniques of plant protection to prevent activation of mycotoxic fungi and to decontaminate the affected crops, there is no comforting solution applicable in Europe yet [9]. This situation highlighted the need for research targeting innovative methods of utilizing contaminated crops which turn back resources to circular agriculture instead of discarding them.

Fish meal is usually produced from wild, marine catches and, according to FAO statistics, only 27% comes from by-products [10]. One ton of fish meal requires about 4–5 tonnes of live fish [11]. Statistics from recent years show that almost 70% of the fish meal is used in aquaculture [12]. Research studies show that marine life has declined dramatically in recent decades, partly due to overfishing [13]. Insect meal as an alternative source of protein could replace fish meal [14].

Thus, our experiment studied an innovative treatment of mycotoxin-contaminated crops, using yellow mealworms (*Tenebrio molitor*), to evaluate the possibility of complete fish feed with these insects fed on toxin-contaminated crops. And as the circular approach requires, the by-product of the treatment, the insect frass, was also studied both from analytical and ecotoxicological standpoints, as a part of the evaluation process.

### 1.1. Aflatoxin

Mycotoxins, including AfB1, are a group of toxic compounds produced by certain fungi that can contaminate various food and feed commodities [15]. AfB1 is well-known for its carcinogenic, hepatotoxic, and teratogenic effects [16]. Therefore, several thresholds are set to prevent human and livestock toxicosis. These substances have garnered significant attention due to their adverse effects on animal and human health. The importance of understanding and addressing mycotoxins, particularly AfB1, lies in their potential to cause severe illnesses and contribute to economic losses in agricultural and food sectors [17,18].

The presence of AfB1 in feed can have significant impacts on the feed industry as AfB1 is highly toxic to animals, affecting multiple organs and leading to various health issues. Consumption of feed contaminated with AfB1 can result in reduced feed intake, poor growth, immunosuppression, liver damage, and even death in severe cases [19]. All livestock but poultry are particularly susceptible to aflatoxin toxicity [20]. This can result in decreased productivity, increased mortality rates, and compromised animal welfare which can have significant economic implications for the feed industry [21]. Decreased animal performance, including reduced weight gain and milk production, can lead to financial losses for livestock producers. Additionally, aflatoxin-contaminated feed may need to be discarded or diverted for non-food use, resulting in economic losses for feed manufacturers and the entire supply chain.

The International Agency of Research on Cancer (IARC) considered AfB1 to be human carcinogenic at least five times in the past half century [22]. Scientific research has extensively investigated the impacts of AfB1, highlighting its carcinogenic properties [23] and association with hepatocellular carcinoma (HCC) [24]. Numerous studies have provided evidence linking AfB1 exposure to increased HCC risk, particularly in regions with high levels of contamination in agricultural products. Additionally, AfB1 is known to possess immunosuppressive properties and can impair the immune system’s ability to defend against infections highlighting the potential for increased susceptibility to infectious diseases in individuals exposed to this toxin.

AfB1 levels in feed are regulated by government agencies and international standards to ensure animal and human health (IARC, 1985). Feed industry role players must adhere to these regulations and implement rigorous quality control measures to ensure compliance. Besides the human and animal health issues, failure to meet the required AfB1 limits can lead to legal consequences, product recalls, and cause damage to companies’ reputation. The Hungarian feedstuff regulations are harmonized with the EU regulations which are shown in the following table (Table 1).

### 1.2. Protein Source of Feed Industry

The feed industry plays a critical role in providing protein for livestock and aquaculture. Historically, the primary sources of protein in animal feed have been derived from crops such as soybeans and fish meal and many others like blood, bone, and feather meal in the past. However, it is worth mentioning that the sustainable use of protein sources for animal feed is a subject of ongoing concern and research. Increasing global demand for animal products, coupled with limited land and water resources, has prompted the exploration of alternative protein sources. In recent years, there has been increasing interest in alternative protein ingredients such as algae, single-cell proteins, and insect meal [25,26,27,28]. These alternative sources could have the potential to supplement or even replace traditional protein sources, thereby reducing the reliance on crops like soybeans.

Soybean meal, in particular, has been a widely used protein source due to its high protein content and amino acid profile. It is a valuable ingredient in many types of animal feed, providing essential nutrients for growth and development, but its production is surrounded by several environmental concerns, such as deforestation, land conversion, CO_2_ release, etc. [29].

Alternatively used fish meal is derived from small, oily fish such as anchovies and sardines, and it is valued for its high protein content and essential nutrients. However, there has also been anxiety about the sustainability and availability of fish meal due to several factors. Overfishing and the decline of fish stocks in some regions have raised concerns about the long-term viability of relying heavily on fish meal as a protein source [10]. Moreover, increasing demand for fish meal from aquaculture, as well as other industries (such as pet food production), is limiting availability of this feed component, thus raising base material prices and, indirectly, product costs.

### 1.3. Insects and Mycotoxin

For both managing toxic crops and the demand for a new protein source in the feed industry, insect rearing can be a part of the solution.

Insects such as yellow mealworm (*Tenebrio molitor*) and black soldier fly (*Hermetia illuscens*) show the potential to utilize mycotoxin-contaminated feed according to previous research. According to [30], zearalenone mycotoxin (ZEA) did not affect mortality of *Tenebrio molitor* larvae while the insects were able to gain weight in an 8-week feeding period. They also found that species from the taxonomic order of Coleoptera were less affected by AfB1 exposure, based on mortality changes, than *Hermetia* species [31]. In the scientific literature various results are shown, but beside mortality one of the most important variables is weight gain and how it is changed by mycotoxin intake. Yellow mealworm larvae showed significantly less weight gain due to exposure to wheat bran contaminated by 8000 µg/kg DON in a two-week feeding period [32]. However, the same species fed on grains contaminated by 500 µg/kg OTA or 500 µg/kg T-2 gained more weight than the negative controls [33]. Some other studies showed that a moderate concentration of AfB1 resulted in higher weight gain, while a high concentration did not affect them significantly, compared to control feed [34]

Based on these results we took one step further from the analytical studies and arranged our research with the aim of evaluating whether the utilization of certain mycotoxin-contaminated crops by insects is a possible way of producing feed grade insect meal. Thus, in our study we worked with the larvae of yellow mealworm (*Tenebrio molitor*) because of the relevance of this species in the insect industry and the ease of breeding. The AfB1 mycotoxin was chosen for testing because of its reputation and the fact that, to date, it is the only mycotoxin that is strictly regulated in feed components in our country. Our aim was to utilize crops that highly exceed the AfB1 threshold of 20 µg/kg and convert them to a fish feed component, via insect production. To investigate, the mealworms were fed on the contaminated crop for several weeks, insect meal was prepared from them, and we analyzed their AfB1 content by HPLC MS/MS in an accredited laboratory to quantify whether the quality fits the regulation. Measuring fish toxicity of insect products and residual frass was also a part of our study to verify their applicability in practice.

## 2. Results

AfB1 content of the insect meal was only measurable in the case of C4 larvae groups as all the other samples were under the detection limit. The AfB1 content of frass decreased gradually within the less contaminated feeding groups (Table 2).

The AfB1 concentration in the pooled replicates of frass samples reduced by 57.5% for C1, 82.8% for C2, 80.8% for C3, and 62.1% for C4. The concentration reduction in the insect meal was higher than in the case of frass. The AfB1 concentration of the dried larvae fed contaminated corn was under the detection limit for C1, C2, and C3 while it was 2.2 µg/kg in the case of the highly contaminated C4 group which is close to a tenth of the EU threshold for animal feed.

At the end of the mycotoxin-contaminated feed experiment, larvae mortality and wet and dry weight were measured and dry matter content was calculated which is shown in Table 3. During the whole 35 days a total of two dead larvae were found: one individual in the 1st replicate of the CC group, one in the 2nd replicate of the C3 group. The groups contained around 840 individuals (210 ± 5 replicates), thus larvae mortality was far under 1%.

Physical properties of fish fry showed similar results among feeding groups. Normal, 33% CC, and 33% C4 groups had 230.68 mg, 242.25 mg, and 191.39 mg wet weights and 44.96 mg, 48.04 mg, and 39.2 mg dry weights, respectively. In terms of length, the 33% CC group was the longest with an average of 1.3 cm, while normal and 33% C4 groups both measured 1.2 cm. Average survival of the fish fry at 28 days were respectively 360, 407, and 369 compared to the starting 500 individuals on the first day. Survival percentages were 72.0%, 81.4%, and 73.8% for normal, 33% CC, and 33% C4, respectively (Table 4). (At the end of the fish feeding experiment, in one of the tank pools in the 33% C4 group, the sponge preventing the fish from escaping was moved, so it is possible that some of the fish escaped, thus this pool was excluded from further analysis).

The comparison of three fry feeding groups of normal, 33% CC, and 33% C4 by one-way Welch’s ANOVA showed that the property of wet weight of fish has a *p*-value of 0.1124, with W being 3.121. Dry weight had *p*-value of 0.214, with W being 1.980. Based on these values none of the feed groups is assumed to be significantly different in weight properties (Table 5). For length, the Welch ANOVA yielded a *p*-value of 0.016 and a W value of 7.78, indicating a statistically significant difference. This finding was further supported by the post hoc analysis (Dunnett’s T3 test), which revealed a significant difference between the 33% CC and 33% C4 groups (*p* = 0.0127). No such differences were observed in the comparisons between the normal and 33% CC groups, or between the normal and 33% C4 groups (*p* = 0.5914 and *p* = 0.5806, respectively). For survival rate the Kruskal–Wallis test was used which had a result of 2.1 with a *p*-value of 0.350, which is higher than the 0.05 significance level, indicating that there is also no statistically significant difference in survival percentages between the three groups. The pairwise comparisons using Dunn’s test also validated this outcome. Although the experiment was conducted during one of the most sensitive developmental stages of the fish, the results should be interpreted with caution. In the future, the analysis should be complemented by additional investigations, such as histological or possibly gene expression studies, to be sure no other changes were caused by these toxins for the fish.

During the insect feeding, a significant amount of frass is produced as a by-product, the further fate of which (destruction or use as a more advantageous material for circular farming) is significantly influenced by its toxicological status. Therefore, as a next step, the toxicological quantification of the frass from different contaminated corn fractions was also carried out. As a first step, our goal was to determine an extract concentration that could be used for all samples and that was suitable for comparing samples with different AfB1 concentrations. For its optimization, we selected the frass with medium mycotoxin content (from C3), based on the analytical studies, and prepared aqueous extracts of 250, 125, and 12.5 mg/mL from it. The extracts were injected into the embryos in a volume of 2.8 nanoliters (nL). When choosing the volume, we took into account the need to deliver an easily adjustable volume that would not cause egg trauma to the embryos [35,36]. Based on the mortality results, the two most concentrated extracts caused 100% mortality in the treated embryos five days after injection. The 12.5 mg/mL extract only caused lethality in about half of the embryos (43.34 ± 6.67%), which was considered potentially suitable for comparing samples with both lower and higher AfB1 content (Figure 1). In further experiments, this concentration of extract was prepared from all samples for injections.

The 12.5 mg/mL concentrated extracts of control corn (CC), the frass produced by larvae from it (C0), and the frass produced from the different toxin-containing corns (C1–C4) were injected at four different volumes (0.52, 1.02, 1.76, and 2.8 nL). In general, it can be said that increased injected volume caused higher mortality in the case of AfB1-containing frass extracts (Figure 2). The largest injected droplet size (2.8 nL) was the most suitable for ranking the tested samples in terms of toxicity. In this treatment, the mortality results of the three frass samples with the highest AfB1 content are significantly different from each other (*p* ˂ 0.05). Although the mortality results of the two samples with the lowest mycotoxin content differed (C1 23.34 ± 3.85%; C2 31.67 ± 3.33%), this was not significant (*p* ˂ 0.05). The frass of the most toxic sample (C4) caused mortality of 91.67 ± 3.33% of the zebrafish embryos. The sample with the second highest AfB1 content (C3) also showed high mortality values in embryos treated with 2.8 nL (43.34 ± 6.67%).

There was no significant difference in mortality between the control corn extract (C0) and the non-injected control group, not even in the groups injected with the largest droplet size. The frass (injected with 2.8 nL) originating from larvae fed on control corn (CC) slightly increased the number of dead embryos compared to the control corn extract, but there was no significant difference between the two treatments (*p* ˂ 0.05). Thus, it can be said that the toxicity of the samples was primarily influenced by the AfB1 content of the frass.

In addition to mortality, sublethal endpoints were also analyzed in the case of surviving embryos injected with the 2.8 nL volume after 120 h of exposure. The extracts from corn containing no toxin (C0) and the frass formed from it (CC) did not cause phenotypic changes in the treated embryos compared to the non-injected control. However, sublethal effects of frass extracts (C1–C4), which originated from AfB1-containing larvae feeding experiments, were detected. The most characteristic symptom was uninflated swim bladders, which appeared in all treated embryos. In addition, in the individuals treated with C3 and C4, there was a deformity of the lower jaw and pericardial edema. Deformity of the yolk sac and curvature of the tail region were only seen in embryos injected with the (highly AfB1-contaminated) C4 sample (Figure 3).

According to previous studies zebrafish embryos exposed to AfB1 show characteristic phenotypic changes, e.g., small and not well-defined olfactory regions, moderately bent bodies, mildly wavy dorsal fins, irregular caudal fins, irregularly shaped lower and upper jaws, uninflated swim bladders, and abnormalities in gastrointestinal tract development and in the yolk [37,38]. Therefore, the observed changes were most likely caused by the mycotoxin content of the frass.

## 3. Discussion

We found a small number of dead larvae which overlaps with the findings of [30,34]. The toxin reduction was an expected outcome based on the literature [31,33]. However, toxin content of insect meal was even lower than what [34] shows. The scientific literature has already confirmed several usages of insect products in aquaculture [39], but our results of the fish feeding experiment strengthened the idea that using mealworm meal reared on contaminated corn is a possible option for feeding common carp fries. It should be noted, however, that sublethal effects cannot be ruled out and may only be detectable through long-term studies; therefore, further investigations with extended observation periods are recommended. Residual frass from conventional larvae feeding can be used as a soil amendment for plant growth and resistance to insect herbivory [40]. In these applications plant toxicity of the frass must be measured. But in our study the frass was derived from larvae fed on AfB1-contaminated corn, meaning that from a holistic point of view it was highly important to test toxicity on a higher taxonomic level to understand the effects and limitations of frass application in soil, compost, or aquatic dilutions. Reduction of AfB1 in the frass coincided with the findings of [41] using similar concentrations of AfB1 in the feed. Toxicity tests showed both lethal and sublethal effects of frass extracts originating from AfB1-consuming larvae. Alongside the remaining AfB1 it is assumed that metabolites such as aflatoxicol (AFL) and aflatoxin M1 were also present in the frass. According to [42], aflatoxin M1 is most likely to occur and has similar carcinogenic properties to AfB1 [43]. The present research focuses on the EU-limited parent compound AfB1 and its degradation, however, investigation of metabolites could be one step further for future research.

## 4. Materials and Methods

### 4.1. Applied Mycotoxin-Contaminated Crops

The original corn sample contaminated with AfB1 derived from a laboratory toxin producing experiment executed in our institute (Institute of Aquaculture and Environmental Safety of the Hungarian University of Agriculture and Life Sciences (MATE)), by the following method: 1 kg of corn meal was moistened by 400–400 mL ionized water, sterilized at 120 °C degrees, for 15 min. After the heat treatment the matrix was inoculated by a 20–20 mL 2 optical density (OD) *Aspergillus flavus* Zt80 strain, previously isolated at the institute. The inoculated matrixes were incubated at 28 °C for 24 days. After the incubation, both matrixes were sterilized at 120 °C for 15 min and dried at 60 °C degrees for 48 h. Both of the original samples were analyzed by HPLC MS/MS according to WBSE-97:2014 ISO [44] compatible standard at Eurofins Hungary Ltd. (Budapest, Hungary) to determine the toxin concentrations. The original concentration of AfB1 in the aflatoxic corn was 1324 µg/kg.

Based on the original concentration, four different AfB1 concentrations were prepared by mixing (diluting) the original samples with uncontaminated, clean grains (Table 6). Clean grains of corn originated from a local feed shop. Their AfB1 concentrations were also analyzed by the same method and by the same accredited analytical laboratory. The fifth group was the negative control, exclusively using clean grains. The five different concentration levels were set to be used in samples with AfB1 contents under, above, and highly above the EFSA threshold of 20 µg/kg, and final mycotoxin contents were measured again. In all cases, not only AfB1 but a summed concentration of aflatoxins B1, B2, G1, and G3 were measured (SumAFLs). According to these results the targeted cereals mostly contained AfB1, which is the only one monitored according to government regulations.

### 4.2. Larvae Preparation

*Tenebrio molitor* larvae used in this research originated from our self-derived colony. The mealworm colony is kept on wheat bran and vegetables scraps, in a vertical system, at 22 °C and 50–55% humidity. Pupae and adults are sorted weekly to establish a stable and well-organized colony. The larvae used for the research were sorted by size then portioned out by weight and placed in glass containers of 9 × 9 × 6 cm^2^ (length × width × height). For sorting a sieving screen was used with 1.5 mm wide, linear openings. In each replicate 20 g of larvae was used which contained approximately 210 ± 5 individuals. For each of the five feeding groups four technical replicates of larvae were prepared, as shown in Figure 4.

### 4.3. Larvae Feeding Period

On the first day, after one day of fasting, the residue (called frass) was removed, and the containers were cleaned and marked. To equalize the water and micronutrient intake of replicates 1% pure microbial agar gel was used, that was cut into equal pieces of 5 g and added to each container in weekly doses. The presterilized gel was made from 5 g microbiological agar (BAA10500, Bacteriological Agar, Biolab Zrt., Budapest, Hungary) and 500 mL of tap water. During the feeding period 40 g of feed and 5 g of gel, as a water source, were added to each container and stored at 22 °C with 50–55% humidity. The end of the feeding period was adjusted to pupation: the experiment was finished on the 35th day, when the first group of larvae was detected to reach a pupation rate of 2.5% (5 pupae appeared).

### 4.4. Preparation of Analytical Measurements

By the end of the 35-day feeding period the larvae consumed most of the provided feed, increased their body size, and produced a notable amount of excrement. The excrement of the larvae and the unconsumed feed (frass) was collected from the bottom of the containers at the end of the feeding period. The insects were sorted from the frass by a wire jig and inspected to look for mortality signs. Insects and their frass were stored separately by concentrations and their replicates. Inactivation of the larvae happened during freezing at −20 °C. The larvae of the 4 technical replicates within each group were pooled. Twenty individuals from each pooled group of larvae were measured for wet weight and then dried at 60 °C for 24 h to calculate the dry matter content. The remaining larvae were also dried and ground by a laboratory mortar, then 5 g was sent for analytical measurement. The remaining dried and ground larvae were stored by group in 15 mL Falcon tubes at −20 °C before the fish feeding study. Frass from each of the 4 technical replicates was pooled to create samples and sent to Eurofins Hungary Ltd. together with larvae samples for HPLC MS/MS (Figure 5). The applied method screened the concentrations of aflatoxins B1, B2, G1, and G3 and summed aflatoxin (SumAFLs), the same as for the original feed samples. However, consumption and digestion by the larvae may have resulted in the formation of several aflatoxin metabolites. In our study we focused on verification of our concept about the possibility of using AfB1-contaminated corn for feeding fish which is the technically limiting factor of application. Our primary aim was to carry out a prompt analysis (screening) of indirect toxicity (from contaminated corn to fish via mealworm larvae).

### 4.5. Preparation of Fish Feeding Experiment

A possible application of mealworm larvae, in circular agriculture, is to feed them to aquacultured fish. Our fish feeding experiment aimed to feed rearing stage common carp (*Cyprinus carpio*) fries with the mealworm larvae previously fed an AfB1-contaminated corn diet. The experiment was carried out in a recirculation fish rearing unit in our institute. Pooled and dried *Tenebrio molitor* larvae from the control (CC) and highly AfB1-contaminated-corn-fed larvae groups (C4) were further ground, then sorted into 0.2–0.4 mm particles which is the same size as the generally used fish feed “Infa” (Aller Aqua Polska Sp. z.o.o., Golub-Dobrzyn, Poland).

Ten-day-old carp fries were sorted into 10 L water tanks to include 500 individuals each (50 fish fry/L). Based on prior experiments of ours it was ideal to replace 33% of fish feed by insect meal [39]). Thus, three carp fry groups were formed with different feeding scenarios: one contained 33% insect meal from CC larvae (33% CC group), one contained 33% insect meal from C4 larvae (33% C4 group), plus the negative control (normal group), which contained only the aforementioned general fish feed “Infa”. The experiment was carried out in five replicates of the three mentioned (CC, C4, and normal) groups.

Carp fries were fed *Artemia nauplii* during their first four days of life. This was followed by a three-day feed exchange period, when the fish fries became used to artificial feed. After that the fish feeding experiment lasted for 28 days. Fish were fed ad libitum and were maintained 4 times a day. Water temperature was 25 ± 1 °C and oxygen level was maintained at 6 ± 1 mg/L during the experiment. Total ammonium (NH_3_/NH_4_^+^) was 0.3 ± 0.05 mg and nitrite (NO_2_^−^) was 0.09 ± 0.04 mg/L, while nitrate (NO_3_^−^) level was 13.4 ± 2.4 mg/L. Level of pH was set at 7.9 ± 0.1. At the end of the feeding trial the surviving fish were counted and 20–20 individuals per tank were weighed and measured. It should be noted that one tank had to be excluded from further analysis (thereby reducing the number of replicates and, consequently, the reliability of the results) due to a technical error that led to the escape of a portion of the fish (33% C3 group). The exclusion was based on an outlier test using Grubb’s method (Z = 1.717). The total length of the fish was determined by photographs using ImageJ 1.54 software (developed by National Institutes of Health and the Laboratory for Optical and Computational Instrumentation, University of Wisconsin, US). After measuring wet weights, the fish were placed in a drying cabinet and, according to the OECD standard (OECD 210, Fish, Early-life Stage test), dried at 60 °C for 24 h and then weighed again. A Mettler Toledo AB204-S analytical scale Mettler-Toledo International Inc., Greifensee, Switzerland) with milligram accuracy was used for determining body weights of the carp fries.

#### Applied Statistical Methods for Fish Feeding Experiment

Survival, wet body weight, dry body weight, and length were studied by statistical analysis. Most of the analyzed groups passed the Shapiro–Wilk normality test (Table 7), meaning that the groups can be described as normally distributed, except for survival percentage of the normal group which had a *p*-value of 0.0009. In all other cases the *p*-value was higher than 0.05, assuming the null hypothesis of the values being normally distributed within the group to be highly probable. The normal group had the lowest *p*-values with 0.219 for wet weight, 0.126 for dry weight, and 0.135 for total length. In the Shapiro–Wilk test the 33% CC group had *p*-values of 0.369, 0.326, 0.314, and 0.502 for wet weight, dry weight, total length, and survival percentage, respectively. The 33% C4 group had the highest probability of normal distribution with *p*-values of 0.850, 0.681, 0.683, and 0.770, respectively. Based on the results of normality tests the properties of wet weight, dry weight, and total length were further analyzed with one-way ANOVA. Since survival data did not pass the Shapiro–Wilk normality test, the Kruskal–Wallis test and pairwise comparisons using Tukey’s HSD test were performed.

### 4.6. Zebrafish Embryo Acute Toxicity Assay with Frass

From the point of view of circular agriculture, it is also important to collect data about the risk of the residual materials. Thus, to evaluate the toxicity of frass, a zebrafish acute toxicity assay (embryo injection) was applied, which is a previously published method used for the evaluation of mycotoxin biodegradation [35]. With this experiment, we wanted to find out, using the example of frasses from corn feeding, whether the microinjection method can be used to rank the remaining by-products in terms of toxicity and whether frass from feeding aflatoxin-contaminated grain could potentially be suitable for use as a soil amendment or organic fertilizer. In the case of appropriate settings, the zebrafish microinjection method is particularly suitable for the examination of samples with a high organic content, such as frass. Zebrafish maintenance and egg collection were carried out in accordance with what was previously described by [35]. Briefly, the wild-type AB zebrafish used for the experiments were obtained from a recirculation fish culture system (Techniplast S.p.A., Buguggiate, Italy) operated at the institute. The fish were kept in a controlled environment (14 h light and 10 h dark periods, 25.5 ± 0.5 °C, pH: 7.0 ± 0.2, conductivity 550 ± 50 μS/cm). The fish were fed twice daily with dry feed (ZEBRAFEED, Sparos Lda., Olhao, Portugal) and once daily with live food (*Artemia nauplii*). The day before the experiments, the fish were placed separately in breeding tanks. On the day of the tests, after the lights were turned on in the laboratory, the partitions were removed according to the microinjection rate to ensure a continuous supply of one-cell stage embryos.

Microinjection was conducted as described by the aforementioned publication [35]. Injection volumes were 100 μm in droplet diameter corresponding to an injection volume of 0.52 nL, 125 μm corresponding to 1 nL, 150 μm corresponding to 1.77 nL, and 175 μm corresponding to 2.8 nL for all tested samples. Each microinjection was performed in four replicates (20 eggs per replicate). After 120 h, mortality was determined based on the following lethal endpoints: coagulated embryo, absence of somites, and absence of cardiac function. In the tests, surviving embryos were anesthetized in MS-222 (100 mg/L tricaine-methanesulfonate) solution and then placed in a 4.5% methylcellulose solution containing an anesthetic and oriented on their right side for photography. Brightfield photographs of embryos were taken at × 30 magnification using a Leica M205 FA microscope equipped with a Leica DFC 7000 T camera and Leica Application Suite X software (Leica Microsystems GmbH, Wetzlar, Germany).

#### 4.6.1. Preparation of Aqueous Extracts

Aqueous extracts of the frass samples were prepared by placing 500 mg of the dry sample into a 5 mL centrifuge tube and adding 2 mL of E3 medium. The tubes were shaken for 24 h on an orbital shaker at 150 rpm at room temperature. They were then centrifuged at 4000 rpm for 10 min, and the aliquots were filtered through 0.22 µm syringe filters. The prepared extracts were stored at 4 °C. The 250 mg/mL extracts were found to be highly toxic to the zebrafish embryos. Therefore, additional diluted aqueous extracts (125 mg/mL and 12.5 mg/mL) were prepared from each sample using the same procedure.

#### 4.6.2. Statistical Analysis of Results

GraphPad Prism 10 (GraphPad Software, Inc., Boston, MA, USA) was used for data analysis. Statistical analysis was performed using generalized linear models (GLMs) with a binomial error distribution and logit link function. For 0% and 100% outcomes, a small continuity correction (±0.5 embryos) was applied to enable model convergence. Estimated marginal means (predicted mortality probabilities) with 95% confidence intervals were obtained for each treatment group, and pairwise group comparisons were conducted on the logit scale with Holm adjustment for multiple testing.

## 5. Conclusions

The aim of this research was to evaluate the method of utilizing aflatoxin-contaminated corn by mealworms to produce fish feed ingredients to make fish meal. The circular approach of ours meant that this research focused on several aspects of the studied method. We investigated the possibility of feeding aflatoxic corn to mealworms, then the possibility of using these mealworms as fish feed ingredients and the physiological effects of that. Lastly, we studied the toxicity of the frass remaining from the first step. Results of our study show that yellow mealworm (*Tenebrio molitor*) larvae feeding on AfB1-contaminated corn have the potential to utilize this agriculture product for its original function when it cannot be used in any other legal applications.

Feeding the larvae on corn with 415 µg/kg of AfB1 produced an insect meal that has an AfB1 content of only 2.2 µg/kg. Therefore, it could be used as a feed ingredient according to the EU/EFSA threshold for animal feed ingredients (20 µg/kg).

The second aspect of our research was to verify the application of the larvae, fed on highly AfB1-contaminated corn, as an alternative feed source in aquaculture. According to the statistical analysis we assume that the insect meal, even when derived from highly contaminated groups, did not affect the development of common carp fries significantly in the 28-day-long feeding period. Therefore, we presume that the *Tenebrio molitor* larvae utilizing AfB1-contaminated grains can be used as an ingredient in fish feed. Although the experiment was conducted during one of the most sensitive developmental stages of the fish, the results should be interpreted with caution. In the future, the analysis should be complemented by additional investigations, such as histological or possibly gene expression studies, to be sure no other changes were caused by these toxin contaminations for the fed fish.

The third aspect was about the toxicity of AfB1-contaminated frass. A zebrafish embryo injection method which was previously used to indicate mycotoxin toxicity of samples with a high level of organic compounds was used, as well as analytical measurements. Based on our results AfB1 concentration of frass decreased notably compared to the initial concentration of feed consumed by the larvae. Despite our results showing significant toxicity in the case of the most contaminated frass samples, there is a possibility that the method of feeding insect larvae with mycotoxin-contaminated crops could be used when contaminated grain lots do not exceed the EU threshold (20 µg/kg) by an order of magnitude. On one hand this kind of circular method could utilize a problematic waste of crop production, transformed by insect rearing into a valuable input of aquaculture. On the other hand this method could also replace a relevant amount of the environmentally problematic fish meal.

The utilization of the contaminated grains was successful, and the results could strengthen the current wave of interest in the use of insect-based fish feeds which could hopefully decrease the environmental impact of aquaculture in the future. We hope that our results can provide a piece of evidence for future reconsideration of the law on this topic to provide a legal environment for this application of insects. Our findings are derived from laboratory-scale measurements, so further investigations are needed to consider the upscaling of application volume.

## Figures and Tables

**Figure 1 ijms-26-09851-f001:**
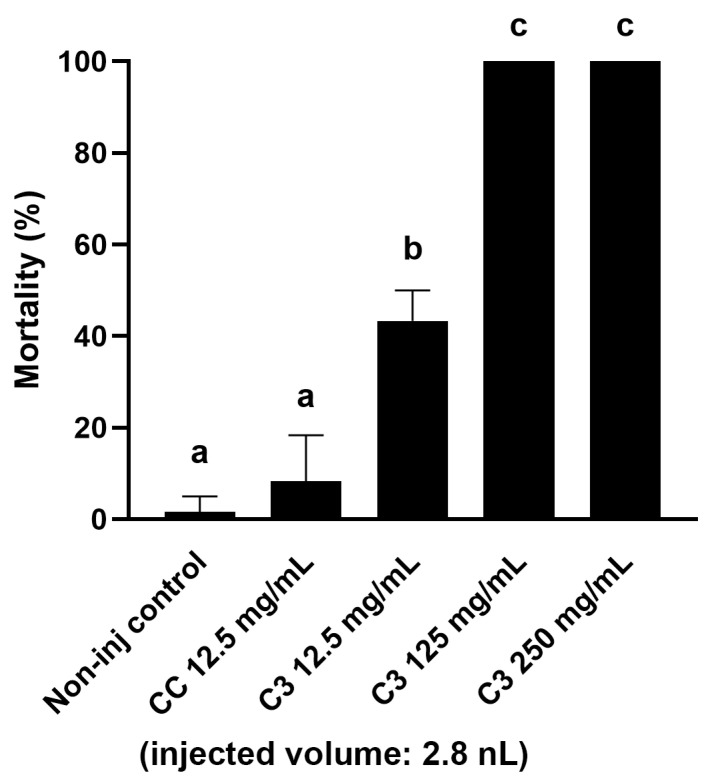
Toxicity of extracts of different applied concentrations made from aqueous extracts of aflatoxin-containing corn-fed larvae frass in an injected volume of 2.8 nL after 120 h of exposure. Bars represent model-estimated mean mortality probabilities (±95% CI) from a binomial GLM. Error bars indicate ± SD. Different letters above bars indicate statistically significant differences between groups (Holm-adjusted pairwise comparisons, *p* < 0.05).

**Figure 2 ijms-26-09851-f002:**
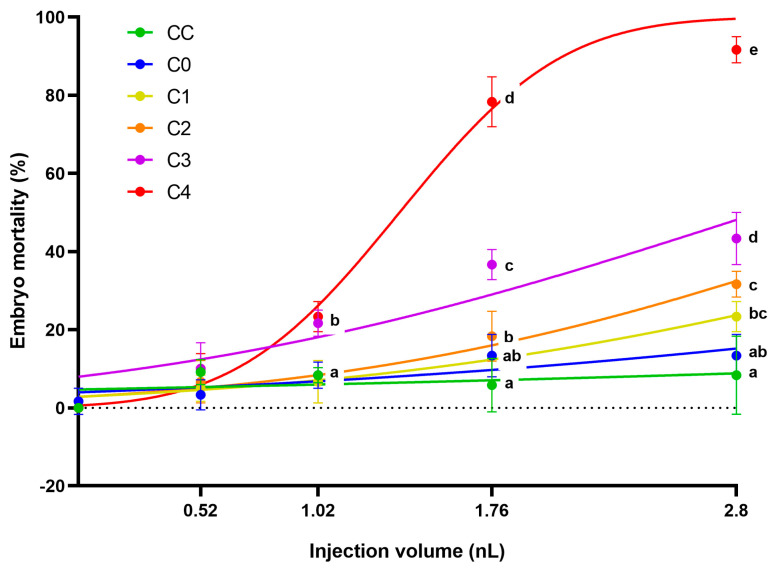
Toxicity (described by mortality rate) of aqueous extracts from the corn feeding experiment of each injected volume after 120 h of exposure. Extract of control corn (C0), extract of frass from control corn feeding group (CC) and from different AfB1 larvae from lowest concentration (C1) to highest (C4). Lines show model-estimated mortality probabilities (±95% CI) from a binomial GLM as a function of nominal concentration. Different letters for each concentration indicate statistically significant differences between treatment groups (Holm-adjusted pairwise comparisons, *p* < 0.05).

**Figure 3 ijms-26-09851-f003:**
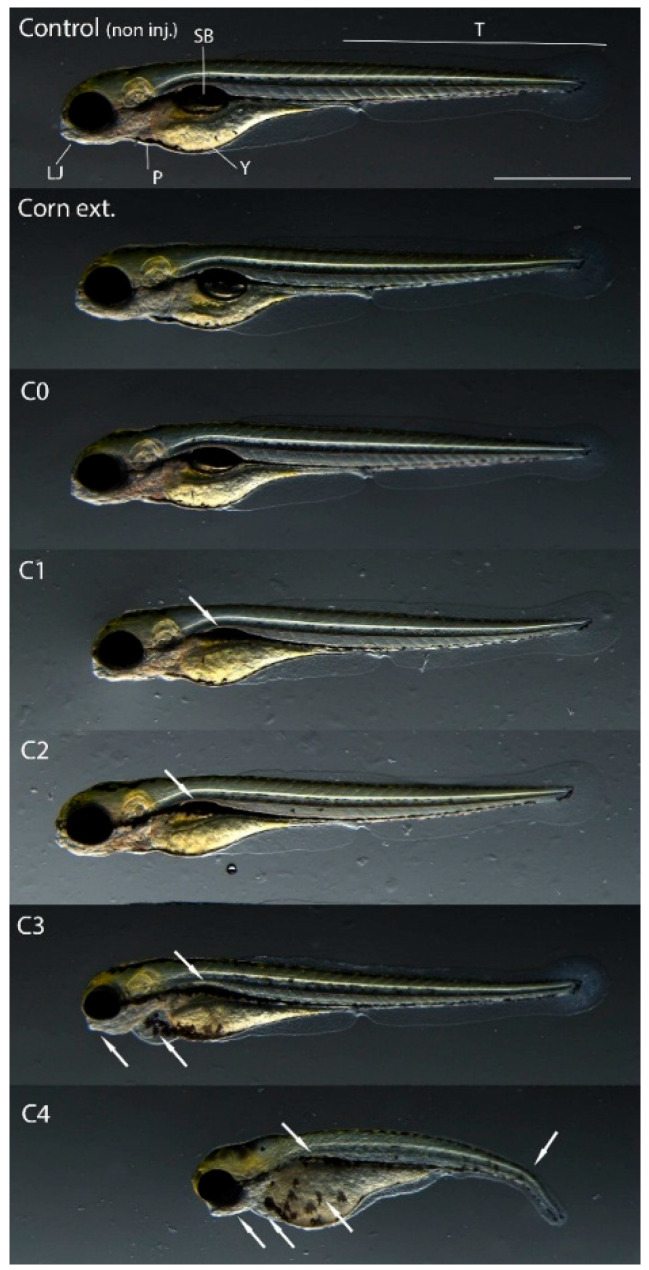
Representative developmental malformations (indicated by white arrows) of surviving embryos injected with 2.8 nL of extracts after 120 h of exposure. Abbreviations: LJ, lower jaw; T, tail; SB, swim bladder; P, pericardium; Y, yolk. Scale bar, 500 μm. Extract of frass from control corn feeding group (C0), lowest (C1) to highest (C4) extracts of frass from different AfB1 concentrations of larvae feeding.

**Figure 4 ijms-26-09851-f004:**
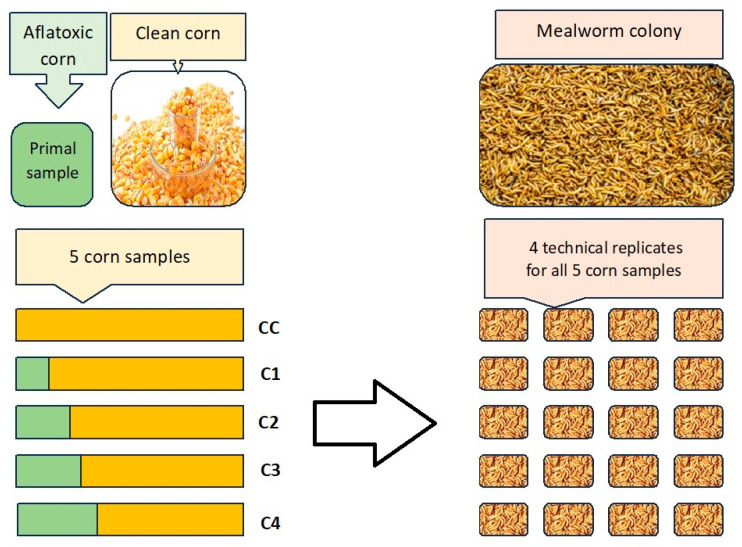
The distribution of 5 corn feeding groups and the structure of 5 × 4 technical replicates of mealworms used in our study. The five samples included one control group of toxin-free (clean) feed and four different concentrations of contaminated feed mixed from the original samples (see Table 1).

**Figure 5 ijms-26-09851-f005:**
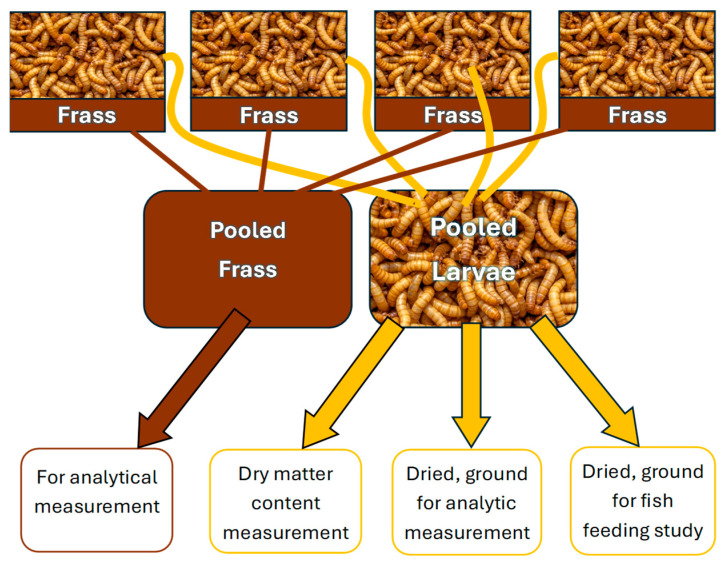
The pooling method of the four different technical replicates within one sample and the later usages of the pooled frass and insect samples.

**Table 1 ijms-26-09851-t001:** AfB1 thresholding concentrations in certain feeds and feed components according to the Directive 2002/32/EC of the European Parliament and of the Council of 7 May 2002 on undesirable substances in animal feed.

Mycotoxin	Maize and Maize Products	Maximum Levels, μg/kg
Aflatoxin B1	All feed materials	20
	Complete feedstuffs for cattle, sheep, and goats with the exception of:	50
	Complete feedstuffs for dairy animals	5
	Complete feedstuffs for calves and lambs	10
	Complete feedstuffs for pigs and poultry (except young animals)	20
	Other complete feedstuffs	10
	Complementary feedstuffs for cattle, sheep, and goats (except complementary feedstuffs for dairy animals, calves, and lambs)	50
	Complementary feedstuffs for pigs and poultry (except young animals)	30
	Other complementary feedstuffs	5

**Table 2 ijms-26-09851-t002:** AfB1 content of larval feed groups, frass, and insect meal according to the HPLC MS/MS analyses. Govt. limit: Representing the EU threshold for feed, detection limit: lowest measurable concentration of the applied analytical method. -: reduction cannot be calculated as values are under limit of detection. Highest AfB1 concentration of corn (C4) is highlighted with orange color fields to express the toxin reduction within feed, frass, and insect meal.

Sample IDs	Aflatoxin B1 Content						
	Concentration (µg/kg)					Reduction (%)	
	Govt. Limit	Detection Limit	Crops	Frass	Insect Meal	In Frass	In Insect Meal
CC	20.0	1.0	<1.0	<1.0	<1.0	-	-
C1			12.0	5.1	<1.0	57.5	>91.7
C2			46.4	8.0	<1.0	82.8	>97.8
C3			90.0	17.3	<1.0	80.8	>98.9
C4			415.3	157.6	2.2	62.1	99.5

**Table 3 ijms-26-09851-t003:** Mortality results and dry matter content of *Tenebrio molitor* larvae after being fed different concentrations of AfB1 in contaminated corn. Dry matter content is calculated from the weights measured before and after drying a portion of the larvae in each group.

Larvae Group IDs	Dead Larvae (pcs/840 pcs)	Weight (g)		Dry Matter Content (%)
		Wet Weight	Dry Weight	
CC	1	8.80	3.59	40.80%
C1	0	12.45	5.05	40.56%
C2	0	10.32	4.12	39.92%
C3	1	11.66	4.67	40.05%
C4	0	10.07	3.99	39.62%

**Table 4 ijms-26-09851-t004:** The average values of physical properties and survival of common carp (*Cyprinus carpio*) fries derived from a 28-day feeding experiment. Three different types of fish feeding group were compared. 1. Normal: contained 100% commercial fish feed as negative control; 33% CC: commercial feed mixed with 33% CC control group insect meal; 33% C4: commercial feed mixed with 33% insect meal from larvae fed with (C4) highly AfB1-contaminated corn.

Group	Wet Weight (mg)		Dry Weight (mg)		Length (cm)		Survival (Piece)		Survival (%)	
	Mean	Deviation	Mean	Deviation	Mean	Deviation	Mean	Deviation	Mean	Deviation
Normal	230.68	85.94	44.96	20.03	1.22	0.10	360	61.12	72.00	12.22
33% CC	242.25	40.48	48.04	8.74	1.28	0.05	407	30.36	81.40	6.07
33% C4	191.39	17.99	39.2	3.77	1.19	0.03	369	40.11	73.80	8.02

**Table 5 ijms-26-09851-t005:** Results of the one-way Welch’s ANOVA tests on different physical features of larvae-fed fish fries. No significant differences were found in the compared features of the three feeding groups of *Cyprinus carpio* fries.

	Welch’s ANOVA (W)	*p*-Value
Wet Weight	3.121	0.1124
Dry Weight	1.980	0.2140
Total Length	7.748	0.0167

**Table 6 ijms-26-09851-t006:** Aflatoxin B1 concentration of five feeding groups of corn measured with HPLC MS/MS. Feed groups are labeled with 2 characters, the first referring to the type of grain (C = Corn, and the second referring to the concentration of the toxin from lowest (1) concentration to the highest (4) and negative control (C)). In the table “SumAFLs” refers to the summed quantity of different measurable aflatoxin types, B1, B2, G1, and G2. “AfB1” refers to aflatoxin concentration B1 exclusively, which is the main targeted aflatoxin type of our study and the only one monitored according to government regulations (EU/EFSA limit value 20 µg/kg). The detection limit of SumAFLs was 4 µg/kg, while the detection limit of AfB1 was 1 µg/kg. Quantities under the detection limit are indicated with “<4.0” or “<1.0”. All samples were measured with HPLC MS/MS by Eurofins Hungary Ltd.

Sample IDs	AfB1	SumAFLs
	µg/kg	
CC	<1.0	<4.0
C1	12.0	16.0
C2	46.4	63.3
C3	90.0	120.0
C4	415.3	616.7

**Table 7 ijms-26-09851-t007:** The results of the Shapiro–Wilk normality test applied on the physical properties and survival data of *Cyprinus carpio* fish fries, measured at the end of a 28-day-long feeding experiment. A *p*-value higher than 0.05 indicates a high probability of normally distributed values of a property within a group.

Property	Carp Fry Group	*p*-Value	Normal Distribution
Wet Weight	Normal	0.219	Yes
Wet Weight	33% CC	0.369	Yes
Wet Weight	33% C4	0.850	Yes
Dry Weight	Normal	0.126	Yes
Dry Weight	33% CC	0.326	Yes
Dry Weight	33% C4	0.681	Yes
Total Length	Normal	0.135	Yes
Total Length	33% CC	0.314	Yes
Total Length	33% C4	0.683	Yes
Survival Percentage	Normal	0.0009	No
Survival Percentage	33% CC	0.502	Yes
Survival Percentage	33% C4	0.770	Yes

## Data Availability

The raw data supporting the conclusions of this article will be made available by the authors on request.

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
