# Peer review of "Utilization of Aflatoxin-B1-Contaminated Corn by Yellow Mealworm Larvae for Common Carp Feed and Assessing Residual Frass Toxicity by Zebrafish Embryo Microinjection"

_ijms, 2025, doi:10.3390/ijms26209851_

Round 1

Reviewer 1 Report

Comments and Suggestions for Authors

Dear Editor-in-Chief, good day

 Date: September 8, 2025

Dear Prof Editor, I am appreciating that you gave me the opportunity to review for your esteemed journal. Upon carefully reviewing this article, I observed that the authors covered a significant topic: Utilization of Aflatoxin B1-Contaminated Corn by Yellow Mealworm Larvae for Common Carp Feed: Assessing Residual Toxicity of the Frass in Zebrafish”. Using insects, the authors aimed to turn crops that significantly above the AfB1 threshold into a component of fishmeal. The authors employed a number of contemporary techniques and multi-species fish serial experiments, along with insects and material from plant origin to achieve ecological and economic benefits. Despite these commendable efforts, the authors should still adequately revise and address some issues that will improve their manuscript. Language needs linguistic improvement. Scientific names are not italicized throughout the document. It is still necessary to improve the discussion and citations to compare the results. Since many paragraphs and sentences are written in a verbose form, they must be rephrased in a precise and scientific way. The conclusion includes references of previous research and is too long. The conclusion needs to be improved to better represent the data and findings of the investigation. Furthermore, I have added remarks to the manuscript. Authors should address and respond adequately.

Thank you,

Author Response

Dear Reviewer #1,

Thank you very much for taking the time to review our manuscript and for your valuable feedback. We greatly appreciate the time and effort you invested. We have carefully considered your comments and suggestions, and we have accepted all of your proposed revisions, which were primarily focused on formatting, improving English grammar, and minor additions in certain sections. We also took your structural criticism into account and have separated the Discussion and Conclusion sections as it was recommended. We really believe your feedback has significantly improved the quality of our paper. All responses of ours are given in the attached .pdf as comment.

Thank you again for your thorough and insightful review!

Reviewer 2 Report

Comments and Suggestions for Authors

General assessment

The manuscript addresses an important applied problem: valorising aflatoxin B1 (AfB1)–contaminated corn through mealworm larvae, assessing its use as fish feed, and testing residual frass toxicity in zebrafish embryos. The experimental concept is relevant and timely. However, there are notable weaknesses in statistical methodology, mechanistic depth, and the strength of interpretation. Below, I organise my comments by manuscript sections.

Note to authors: The PDF version I received does not contain line numbers, which made it difficult to reference exact sentences. For clarity, I have organised my review comments under manuscript section titles and subtitles instead of line-specific references.

Title
The current title is: “Utilization of Aflatoxin B1-Contaminated Corn by Yellow Mealworm Larvae for Common Carp Feed: Assessing Residual Toxicity of the Frass in Zebrafish.”

This partly reflects the work, but it can mislead readers. It suggests zebrafish were fed the mealworm products, whereas in fact zebrafish embryos were only used as a model to test frass extracts by microinjection. To avoid confusion, a clearer title might be:

“…Assessing Residual Frass Toxicity via Zebrafish Embryo Microinjection”

This makes it explicit that zebrafish were only used for toxicity testing of the by-product, not as a feeding model.

Minor comments

Abstract: grammatical correction “does not had” → “did not have.”

Introduction: correct typo “Countles” → “Countless.”

Standardise decimal separators (comma vs. period) and AfB1 notation throughout.

Figures 1–3: improve clarity of axis labels, scales, and units.

Major Comments

Materials and Methods / Statistical Analysis

Wrong post-hoc after Kruskal–Wallis. The authors report Tukey’s HSD after Kruskal–Wallis for fish survival data. Tukey’s is a parametric post-hoc test and not suitable here. Dunn’s or Conover/Steel–Dwass with multiplicity correction would be appropriate.

Assumptions beyond normality not checked. Shapiro-Wilk is reported, but no tests of homogeneity of variance (Levene, Brown-Forsythe). With small group sizes, unequal variances could bias ANOVA results. Welch ANOVA would be safer if the variances differ.

Experimental unit in fish trial. In the carp experiment, the tank, not the individual fish, is the true replicate. One tank was excluded​ due to an “escape event.” Exclusion criteria should be pre-specified, and ideally, a sensitivity analysis including the excluded tank should be presented.

Zebrafish mortality data type. Mortality is binomial (dead/total). Two-way ANOVA on percentages is not ideal. A GLM/GLMM with a binomial link, or beta-regression on proportions with replicate as a random effect, would be preferable. If ANOVA is kept, transformation and justification are needed. The factors used (volume × sample) and whether interaction was tested should be clarified.

Multiplicity control in zebrafish tests. Letters in Figs 1–2 imply multiple pairwise contrasts, but the exact multiplicity correction is not described. Please specify the procedure (Tukey, Dunn with Holm, etc.).

Results and Discussion

Pooling of replicates. For chemical analysis, technical replicates were pooled, meaning variance information is lost. Yet percentage reductions in AfB1 are reported as precise values. This should be acknowledged as a limitation.

Mechanistic interpretation is shallow. Discussion does not explore the metabolic fate of AfB1 in mealworms (e.g. AFM1, AFL metabolites) or the possibility of transformation products in frass. For a molecular sciences journal, more depth is expected.

Overinterpretation of the fish trial. The carp fry endpoints were only survival, wet/dry weight, and length. The Discussion interprets non-significant differences as evidence of “no effect,” whereas the data only show no gross differences under limited power. Sublethal endpoints (biochemical, histological) were not assessed. This limitation should be acknowledged.

Speculative claims. The statement that the method is “most likely usable with no relevant risk when contaminated grain lots don’t exceed the EU threshold by an order of magnitude” is not supported by the data. Only one highly contaminated level was tested. Similarly, speculation about regulatory reconsideration and industrial scaling goes beyond the evidence. These points should be toned down.

Conclusion

The main supported conclusions are: mealworms reduce AfB1, insect meal remains below EU thresholds, carp fry showed no significant gross differences, and frass remains toxic at high contamination levels.

​Broader claims of safety, regulatory readiness, and “no relevant risk” at <10× EU thresholds are over-extensions. These should be reframed as hypotheses or potential implications, not as demonstrated findings.

Author Response

Dear Reviewer #2

Thank you for your valuable time and effort in reviewing our manuscript! We sincerely appreciate your constructive feedback and detailed suggestions. We have carefully addressed each of your comments and suggestions. We are particularly grateful for your excellent recommendations regarding the statistical analysis, which we have incorporated into the revised manuscript to the best of our ability. Furthermore, we have followed your advice to separate the Discussion and Conclusion sections, which we believe has significantly improved the paper's structure and clarity. Thank you again for your insightful review. Your feedback has been instrumental in improving the quality of our work.

All responses to the review can be found under the relevant sections in details in the attached document.

Sincerely yours

Round 2

Reviewer 2 Report

Comments and Suggestions for Authors

Thank you for your thorough and thoughtful revision of the manuscript. I appreciate the considerable effort you put into addressing the comments from the previous round. The manuscript has improved substantially in clarity, methodological robustness, and interpretative caution. Below, I summarise my evaluation of the revised version and offer a few final remarks for consideration:

You have appropriately corrected several key statistical issues. The use of Dunn’s test following the Kruskal–Wallis test is now suitable for the survival data, and re-analysis of the zebrafish mortality with a binomial GLM and Holm correction significantly improves the rigour of the results. Likewise, the use of Welch’s ANOVA, where variance heterogeneity was possible, is a welcome change. These adjustments greatly strengthen the reliability of your conclusions.

I appreciate the clarification regarding the tank-level replication in the carp trial and the explanation of how the outlier tank was handled. While pre-specifying exclusion criteria or presenting a sensitivity analysis including the excluded tank would have been ideal, the justification provided is transparent and acceptable as presented. I encourage you to mention this point explicitly as a limitation in the Methods or Discussion so that readers understand how the exclusion was determined.

Thank you for acknowledging that pooling technical replicates in the chemical analyses results in a loss of variance information. This clarification is important and should remain clearly stated so that readers interpret the precision of the reported reduction values appropriately.

The revised statistical approach to the zebrafish embryo mortality data is a significant improvement. The application of a GLM with a binomial error distribution and proper multiplicity correction is now appropriate for the data type, and the explanation of the analytical approach is clearer. The improved figure clarity is also appreciated.

While I understand your decision to focus exclusively on AfB1 as the regulated parameter, the Discussion still remains relatively shallow from a mechanistic perspective. At a minimum, I would recommend explicitly acknowledging the likelihood of metabolic transformation products (such as AFM1 or AFL) and stating that their identification was beyond the scope of this study. This would strengthen the manuscript’s scientific framing and align it more closely with the expectations of a molecular sciences audience.

The revised text more appropriately cautions against overinterpreting non-significant differences in the carp fry endpoints. The addition of comments about the need for histological or gene expression analyses is welcome. It would be helpful to reiterate in the Discussion that the absence of gross differences should not be interpreted as definitive evidence of no sublethal effects.

The tone of the conclusions is now more balanced, with speculative statements about regulatory implications and safety thresholds appropriately softened. The phrasing of such points as potential future directions rather than demonstrated findings is an improvement.

The revised title more accurately reflects the content and scope of the study. Improvements in figure labelling and consistency of notation also contribute to a clearer manuscript.

With the revisions you have implemented, the manuscript is now considerably stronger and scientifically sound. My remaining suggestions are minor and primarily aimed at improving transparency and framing: explicitly note the tank exclusion as a limitation; briefly acknowledge the potential role of AfB1 metabolites; & reiterate the interpretative limits of the carp fry results.

With these refinements, I believe the manuscript will be ready for publication.

Author Response

Dear Honoured Reviewer,

Thank you again for taking the time to rereview our manuscript and for your valuable feedback! We greatly appreciate the your kind notes about the improvement of our manuscript. We have carefully considered your comments and suggestions, and tried to further improve in all three sections of our paper according to your proposed revisions, particularly in

a.) to note the tank exclusion as a limitation,

b.) to briefly acknowledge the potential role of AfB1 metabolites and

c.) to reiterate the interpretative limits of the carp fry results.

We also made some minor corrections in the text made a double check on References. Some of the referenses were inadequately given thus we also corrected them respectively. Your feedback has significantly improved the quality of our paper. Thank you again for your valuable advises and comments!

Sincerely

The authors
